# IOTA-BT: A P2P File-Sharing System Based on IOTA

**Li-Yuan Hou, Tsung-Yi Tang and Tyng-Yeu Liang ***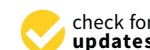

Department of Electrical Engineering, National Kaohsiung University of Science and Technology, Kaohsiung 807618, Taiwan; F107154131@nkust.edu.tw (L.-Y.H.); vazontang@gmail.com (T.-Y.T.)
**\*** Correspondence: lty@mail.ee.nkust.edu.tw; Tel.: +886-7-3814526 (ext. 15508)

**Abstract:** BitTorrent (BT) is the most popular peer-to-peer file-sharing system. According to official BT information, more than 100 million active users use BT for file transfers every month. However, BT mainly relies on either a central tracker (tracker) or distributed hash table (DHT) for locating file seeders while it runs a risk of a single point of failure or cyber-attacks such as Sybil and Eclipses. To attack this problem, we proposed a Peer-to-Peer (P2P) file-sharing system called IOTA-BT by integrating BitTorrent with IOTA in this paper. The advantages of IOTA over blockchain include scalability for high throughput, compatibility with Internet of Things (IoT) footprints, zero transaction fees, partition-tolerant, and quantum-resistant cryptography. The autopeering and neighbor selection of the Coordicide of IOTA is aimed at defending a Sybil or Eclipse attack. IOTA-BT inherits these advantages from IOTA. Moreover, our experimental results have shown that the cost of executing BT functions, such as releasing torrent files and retrieving seeder information on IOTA-BT, is acceptable for improving the security of BT. IOTA-BT can indeed efficiently provide users with a P2P file-sharing environment of higher security.

**Keywords:** P2P; file sharing; BitTorrent; blockchain; IOTA

## 1. Introduction

Peer-to-Peer (P2P) file sharing [1] is a digital content sharing technology based on P2P networks, allowing users to cost-effectively share books, music, movies, and software files. Without any doubt, BitTorrent [2] is the most well-known and used P2P file-sharing system. According to official BitTorrent (BT) information, more than 100 million active users use BT for file transfers every month. When BT was launched in 2001, it used a tracker to provide the seeder information for clients to download their desiring file from the file seeders. Currently, BT usually applies a Distributed Hash Table (DHT) [3] to distribute the information of file seeders. By using a DHT, users obtain the location of seed nodes that own the file without a centralized tracker. Although the previous two methods are useful for clients to acquire information about file seeders, the central server has a single point of failure. When the tracker fails or shuts down, the BT client becomes unable to see the information of file seeders. In contrast, a DHT is vulnerable to cyber-attacks, such as Sybil [4], Eclipse, and Pollution [5]. With the consideration of system security, we try to develop a P2P file-sharing system based on the blockchain technology in this paper.

Since Satoshi Nakamoto published BitCoin [6] in 2008, blockchain has drawn lots of attention from researchers. Not only does it use decentralized nodes to maintain the network to achieve decentralization, but it also uses a miner verification mechanism to generate tokens. Consequently, intangible assets in the virtual world can be brought into the real world to achieve a substantial asset transfer. BitCoin is undoubtedly the pioneer of virtual currency, allowing the world to see the potential of blockchain. Therefore, blockchain has become a hot research topic, and many blockchain-based applications, such as system design [7], product traceability [8], crowdsourcing [9],

energy markets [10], education credit [11], edge computing [12], and vehicular networks [13], have been proposed in recent years.

In the Ethereum [14] network, a DHT is usually used as a selection mechanism of peer nodes. However, the maintenance of the blockchain network is Proof-of-Work (PoW) by miners. Although malicious nodes can enter and exit the network arbitrarily, they must make their computing power exceed 51% of the entire network if they want to launch an attack. Additionally, it is time-consuming to allow subsequent transactions to be attached to the illegal chain to make the tampered content valid. Therefore, malicious network attacks on the blockchain are not cheap. The consensus mechanism successfully maintains the security of the blockchain network. There have been many studies [15–17] using blockchain technology in related file storage applications to achieve access control, file ownership, and file originality. Although PoW can maintain the security of the entire blockchain, any transaction requires miners to verify, and it takes a certain amount of time to perform operations to verify the transaction, which makes blockchain transactions inefficient. To resolve this problem, some alternative ways of substantially speeding up blockchain transaction throughputs have been proposed in past research.

For example, the Lightning network [18] is a second layer technology used for improving the transaction speed of blockchains. The second layer consists of multiple payment channels between two users. Using the channels, the users can make or receive payments from each other. The transactions in the channels are updated to the blockchain only when two users open and close a channel. However, a failure at one lightning node may easily crash the entire network. Besides, lightning networks are vulnerable to cyber-attacks because they are required to be online at all times. RapidChain [19] is a solution proposed to solve the bottleneck of low transactions per second (TPS), low scalability, and low storage capacity of BitCoin. The consensus mechanism in the RapidChain system is to form a reference committee by nodes and introduce the concept of epoch to iterate the members of the reference committee. Since the routing protocol between the client and the committee refers to Kad (Kademlia), the composition of the committee is continuously updated to avoid Sybil attacks. Polkadot [20] is aimed at creating a network protocol to solve the isolation of the blockchain. By establishing a multi-chain architecture, all different blockchains connected to this architecture can exchange information. To achieve this goal, Polkadot uses Parachain for data calculation and transaction information processing. Through multiple parallel chains, the horizontal expansion of the blockchain can be achieved to improve the performance of blockchains. On the other hand, Polkadot uses Relaychain to verify the blocks given by each Parachain and provide proof of finality. Omnilayer [21] is a software layer built on top of BitCoin, which provides users with a distributed and peer-to-peer platform for trading. The advantage of Omnilayer is its ability to support user-generated currencies and smart property while its drawback is that it is inherently vulnerable to malicious attacks. Although the previous distributed technologies are useful for improving the performance of blockchains, they are dependent on miners and need pay fees for transaction proof. Therefore, they are not cost-effective for a voluntary file-sharing environment such as BitTorrent.

Fortunately, the emergence of IOTA [22] has dramatically improved the efficiency of blockchain transaction verification. It abandoned the chain structure of the previous blockchain and used DAG (Directed Acyclic Graph) to form its Tangle network. Whenever users need to post a transaction, they only need to verify two past transactions to complete the transaction process, which significantly improves the throughput and scalability of the blockchain. More important is that IOTA supports zero transaction fees and does not rely on miners for transaction proof. Therefore, IOTA has been widely used in the Internet of Things. For example, the IOTA Data Market [23] cooperates with many well-known companies for secure storage and transactions. Bilal Shabandri [24] et al. proposed publishing data collected by IoT devices (such as smart meters and smart charging systems) to the Tangle network for enhancing privacy and security.

As previously described, we propose an IOTA-based BitTorrent system called IOTA-BT for P2P file sharing in this paper. By replacing the central tracker, it solves a single point of failure and avoids

the problem of DHTs being vulnerable to cyber-attacks because it inherits the advantages of IOTA. When a BT client attempts to be the seeder of a file, it publishes the seeder information to the IOTA network. When other clients want to download the file, they can query the transaction content on any IOTA node, and then know the seeder and file information. Therefore, the proposed system supports the distributed query and ensures the accuracy of information. In addition, this research tries to add a timestamp when BT clients are seeding so that later other clients can quickly find the latest information about the seeder nodes.

The remaining chapters of this paper are organized as follows. Section 2 introduces relevant background knowledge. Section 3 describes the framework of IOTA-BT. Section 4 discusses the efficiency of the proposed system. Section 5 presents the conclusions of this paper and our future work.

## 2. Background

### 2.1. BitTorrent

The communication of the BT network is mainly dependent on HTTP, and the transmission content is mainly the information recorded in the torrent file. The text encoding of the seed file adopts Bencode. The main content consists of the file name (name), the URL list of the tracker (announce), theSHA-1hash value of the file information (infoHash), the path and capacity of a single file, the description of the name (file), the size of all files (length), and the size of each file segment (pieceLength). Whenever a client wants to download a file on the BT network, the seed file of this file must be obtained first. Based on the information recorded in the seed file, the client can determine if it currently owns some of the blocks of the file or not and then request the remaining blocks from the reverse node, and finally, combine all the blocks into a complete file.

The original BT network is mainly composed of a peer and tracker. The tracker belongs to the client–server architecture. It is responsible for maintaining the seeder lists of files. When a peer requests the tracker, the tracker returns the seeder list to the peer for subsequent node communication and file transfer. Peer is a client node that has a unique peer ID with a length of 20 bytes. A peer can download and seed a specific file to become the seeder of this file. If a peer wants to know other peers that own a specified file, it must rely on the tracker's assistance to know the information of the seeders that have the required file.

The basic seeding process is shown in Figure 1. After a client generates a seed file for a specified file, it creates a connection request to the tracker and sends the peer ID, IP, port, and status of its node to the tracker. Then, the tracker creates a peer list according to its corresponding infoHash, which records the received node information, and then returns the peer list and the interval parameters of the heartbeat to the peer. After each heartbeat interval time is reached, the client must send node information again to let the tracker confirm that the client is still alive. Besides, it waits for file requests from other clients for the rest of the time.

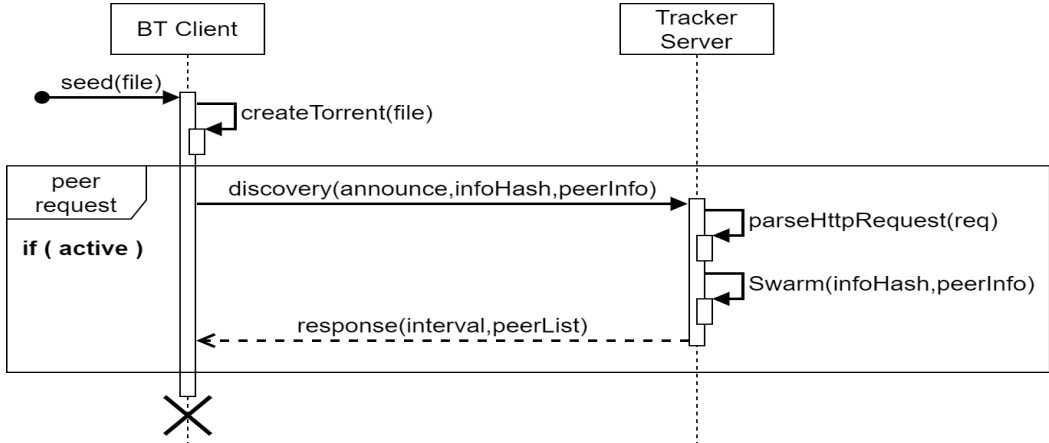

**Figure 1.** The seeding process of BitTorrent (BT).

On the other hand, the file download process is shown in Figure 2. A client must use the seed file to send a request to the tracker recorded in the file content. After receiving the seeder list and heartbeat parameters, it sends the file request to the seeders recorded in the peer list and waits for the seeders to reply and receive the file partitions coming from the seeders. The client confirms if the file partitions are the same as those recorded in the seed file to ensure the integrity of the file. After receiving all the file partitions, the client can assemble these file partitions to form the complete file. At the same time, it can choose to become a seed of the file to serve other clients that need the same file.

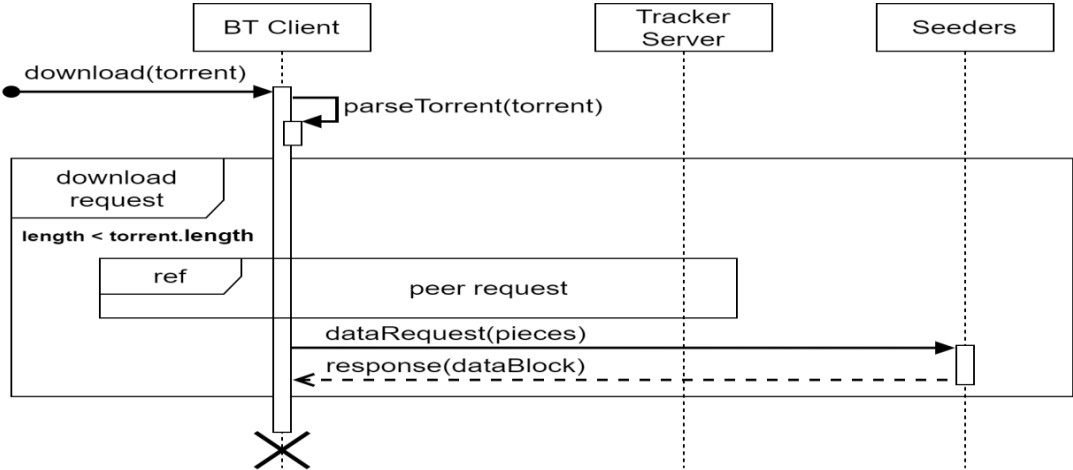

**Figure 2.** The download process of BT.

Because the tracker has the risk of a single point of failure, it cannot stably provide the quality of file transmission. Therefore, the current BT network adopts a DHT (Distributed Hash Table) for node information exchange. DHTs discard the centralized server and hashes the nodes and data into values. The value of the node is its node number in the DHT network, and the value of the data is the target node for storage. By dispersing the information in the nodes of the DHT network, the single point of failure problem generated by the centralized server is solved. BT's DHT network uses Kademlia (Kad) [25]. The contents of the hash list stored in Kad are described as follows. (1) Keywords: it is used to query the corresponding file name or information. Performing SHA-1 hashing of the keywords to obtain a result with 160 bits as the key, the corresponding result is the file message list. (2) File index: according to the query result, the node that owns a file is found. The SHA-1 hash value of the file is used as the key. The corresponding value is the information of the node. In Kad, the node length is 160 bits, the same as peer ID. The distance between nodes is not based on the physical distance or the number of routers, but the XOR method is used for judgment. Assume that the two nodes are a and b, respectively. The distance d is equal to a XOR b. The smaller the value of d is, the closer the nodes are. Kad represents the key of the keyword, file index, and node ID by 160 bits to simplify the query format so that any query uses the same format input to achieve the desired purpose.

In the past decade, some research has been conducted for the improvement and application of BitTorrent. For example, F. Costa et al. [26] applied the data distribution technique of BitTorrent to the BOINC middleware. Their goal was to decentralize BOINC's data model to take advantage of client network capabilities. Michael Burford [27] proposed using HTTP or FTP servers as seeds for BitTorrent downloads. As this is supported in widespread clients, seeding for a BitTorrent download could be conducted entirely with a company or personal HTTP/FTP servers. Florian Adamsky et al. [28] demonstrated that the BitTorrent protocol family is vulnerable to distributed reflective denial-of-service (DRDoS) attacks. Their experiments reveal that an attacker can exploit BitTorrent peers to raise network traffic dramatically. Additionally, they observed that the most popular BitTorrent clients are the most vulnerable ones. A. Bhakuni et al. [29] proposed a detection-cum-punishment mechanism to detect and punish free-riders in the P2P network. Their experimental results prove that the proposed

punishment mechanism improves the performance of the P2P network by decreasing the download times of non-free-rider peers of the network and punishes free-riders by increasing their download times. R. Bindal et al. [30] proposed a biased neighbor selection approach to enhance BitTorrent traffic locality. The simulation result showed that the proposed approach maintains the nearly optimal performance of BitTorrent in a variety of environments and reduces the cross-ISP traffic.

On the other hand, the IPFS (InterPlanetary File System) [31] recently received much attention from researchers. IPFS is a protocol and peer-to-peer network that combines the four mature technologies, including BitTorrent, Git, DHT, and SFS (self-certified naming) technology. It allows participants in the network to communicate with each other for storing, requesting, and transmitting verifiable data. Its operating principle is to divide files into blocks to be hashed and save them in blocks to avoid duplicate files on the Internet. However, the files in IPFS must be broadcasted to avoid file-access failure after a single point of offline. In contrast, BT nodes can freely enter and exit the network, share, and download files. Since IPFS also uses a DHT for locating which node has the target file, it faces the same security problem existing in BT.

### 2.2. IOTA

IOTA is a decentralized ledger technology proposed by David Sønstebø, Sergey Ivancheglo, Dominik Schiener, and Dr. Serguei Popov in 2015. It is dedicated to the Internet of Things and Machine-to-Machine (M2M) communication and is also applicable to P2P file sharing. Compared with BitCoin and Ethereum, IOTA uses Tangle to store transactions, as shown in Figure 3.

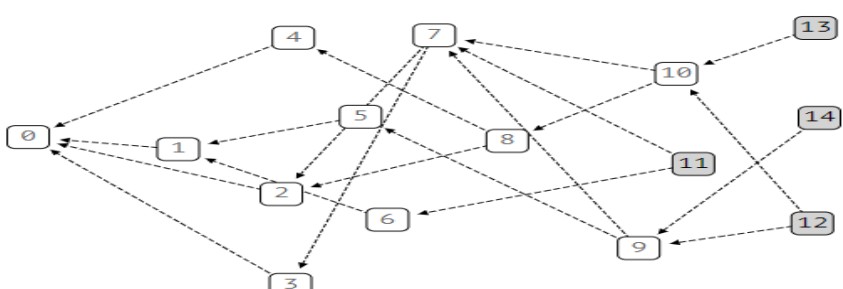

**Figure 3.** Tangle network.

In the Tangle network, except for the first genesis transaction (genesis), the rest of the transactions are initiated using the Markov Chain Monte Carlo (MCMC) algorithm to select two transactions and verify if they are legal. Since IOTA needs to verify the other two transactions when initiating a transaction, the more transactions initiated per unit time, the higher the throughput of transaction verification, thereby eliminating the dependence of the distributed ledger on miners. IOTA transactions (TX) can be divided into Input TX and Output TX. Input TX is used to withdraw money from the address and cannot be a zero-value transaction. Output TX can be used to collect payments or record data and can be valued or zero-value transactions. Several transactions are made into a bundle and checked by the IOTA nodes. The sum of value in a bundle must be 0 to prove that the transaction is a break-even. Moreover, all transactions in the bundle are either accepted or rejected. After the bundle is accepted, the internally recorded transactions are added to the Tangle ledger.

In terms of implementation, IOTA reference implementation (IRI) has been proposed, which is a set of JAVA open-source software. The client can join the IOTA network as a node by executing IRI. The main functions of IRI are (1) verifying transactions, (2) storing and disseminating transactions, and (3) interacting with the client library. In message transmission, IRI uses the gossip protocol to disseminate transactions. When the IRI node receives the transaction sent by the neighbor node, it compares its database. If the transaction is not stored in the database, it is verified and stored in the database, and then is forwarded to other neighbors.

When the client wants to initiate a transaction, the process is described as follows. (1) Prepare Input and Output TX and add them to the bundle. The sum of the bundle value must be zero. (2) Hash the address, value, tag, timestamp, currentIndex, lastIndex, and other information in all TXs through the sponge function to generate a bundle hash and fill the bundle hash into the TX. (3) Use the bundle hash and private key to generate the signature of Input TX and fill the signature into Input TX. (4) Use API's getTransactionsToApprove to obtain Tips from IRI and fill in trunk and branch in each TX. (5) Perform PoW for each TX in the bundle and fill the hash into nonce. (6) Send the bundle to IRI for verification, storage, and broadcasting to complete the transaction initiation. On the other hand, the transaction query process is as follows: (1) Call the findTransaction function by the hash values of bundle, address, tags, and approves to return the hash value of the transaction. (2) Call the getBundle function to view the detailed transaction content by parsing the hash value.

To prove the ownership of the message and prevent interference from spam, IOTA uses the framework of MAM (Masked Authenticated Message). The basic principle of MAM is to use the cotyledons of the Merkle hash tree to generate a signature. The root generates an address. The signature prevents the sender from being impersonated. The root is used to calculate the address where the message is sent and record the root of the next generation hash tree. Through the generational alternation of the hash tree, IOTA prevents sending messages to the same address for a long time to avoid spam attacks. According to the white paper of Coordicide [32], IOTA adopts a Sybil protection mechanism based on mana that is regarded as a hard to obtain resource as well as a form of a reputation, which can be assigned to trustworthy nodes. As mana is credited as part of regular transactions, nodes need not always use their account's private keys to sign, and then avoid a severe security risk. Besides, IOTA uses an autopeering mechanism to make new nodes easily join the network and avoid an attacker targeting specific nodes during the peering process. On the other hand, Bitinfocharts [33] and Etherscan [34] reveal that the block time is 13.2 s and the value of TPS is 12.5 in the Ethereum network. If a client intends to add a transaction into the blockchain in two minutes, it must spend 150 Gwei, which is about USD 1.13. In contrast, the client in the IOTA mainnet adds a transaction bundle into the Tangle only by verifying another two bundles, and it need not pay anything. The IOTA 1.5 office report depicts the value of TPS as over 1000. IOTA is more efficient and economical than Ethereum for clients. As previously discussed, this study exploits IOTA instead of Ethereum to implement the BT framework based on the distributed ledger technology.

## 3. IOTA-BT

The architecture of IOTA-BT is shown in Figure 4. It uses the original BT tracker mechanism but discards the previous centralized settings and runs the tracker on each client. The main functions of the local tracker include packaging node information into transactions and sending them to IOTA for storage, searching for transactions containing node information, assisting the client in disseminating and collecting node information, seeding and downloading.

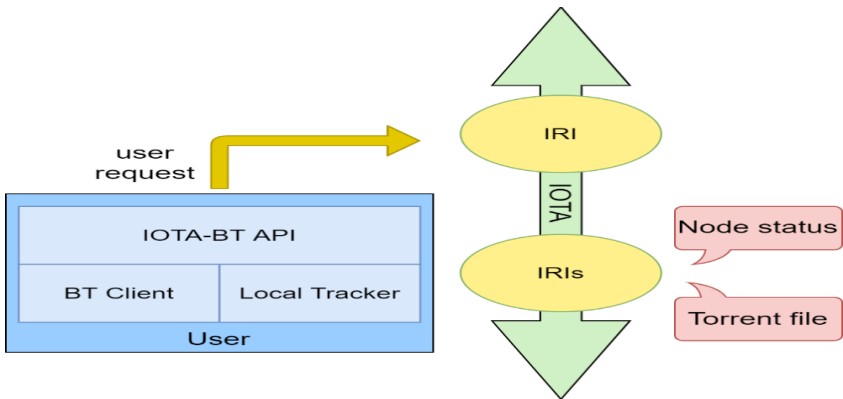

**Figure 4.** IOTA-BT architecture.

Client-side operations include creating the torrent file, seeding, and downloading. In IOTA-BT, file publishers must store their seed files in IOTA, as shown in Figure 5. The process is described as follows: (1) Generate a seed file for the specified file through BT Client. (2) Pack the seed file and time stamp with the MAM message and enter the transaction initiation process. (3) Combine transactions into bundles through the prepareTransfers function. (4) Use getTransactionsToApprove to request IRI, perform tip select to verify two pieces of data and return the TX hash of the two transactions. (5) Send the transaction and TX hash to the PoW node for proof of workload by attachToTangle. (6) Finally, the transaction is stored and broadcasted by IRI through storeTransactions and broadcastTransactions. After completing the publishing process, the client can start file seeding and become the seeder for file sharing. Users can use the root to perform the MAM transaction query and can acquire the detail and release time of the seed file.

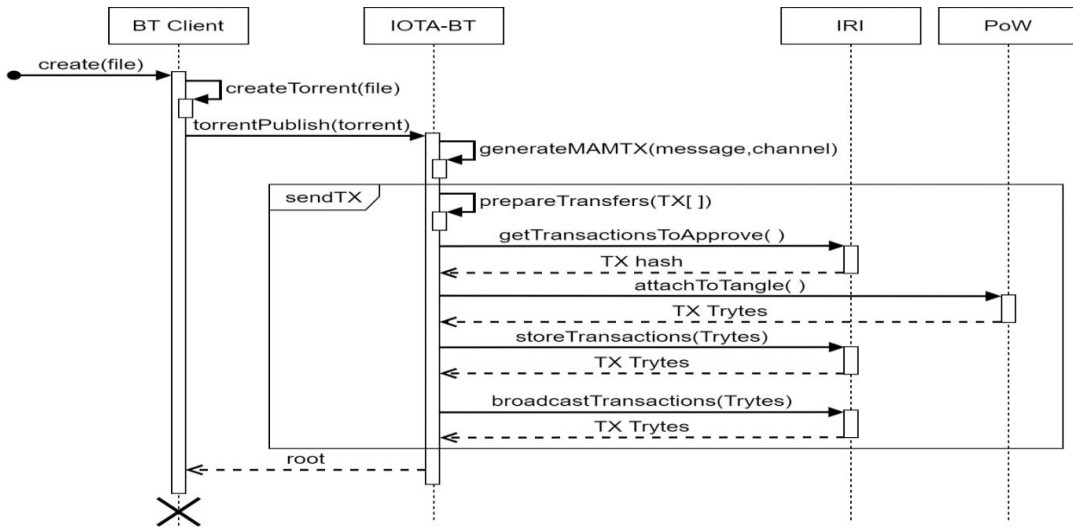

**Figure 5.** Publishing process in IOTA-BT.

To successfully share files with other users, the file owner must seed the files and become a seeder for transmitting the file to the users in need. The seeding process of IOTA-BT is shown in Figure 6. The seeder in IOTA-BT must have the entire shared file and the MAM Root when the file is released.

The seeding process consists of the following steps: (1) The client generates torrent for a given file and obtains the infoHash of the file. (2) Send infoHash, root, and node information to the local tracker. (3) After receiving the request analysis, the local tracker uses the root to query the MAM message to obtain the timestamp and the seed file. (4) Generate the designated address for sending the transaction through generateAddress. (5) Record the own node information in the transaction and send the transaction. (6) Query all transactions under the specified transaction address and obtain all transaction hash. (7) Perform transaction hash decoding to obtain the bundle list and return the transactions in the bundle. (8) After receiving the list of nodes recorded in the transaction, the local tracker creates a peer list of the specified infoHash in its database (Swarm), records all node information in the list, and returns the peer list to the client. As previously described, clients do not need to download and scan the entire Tangle to find transactions of interest related to IOTA-BT.

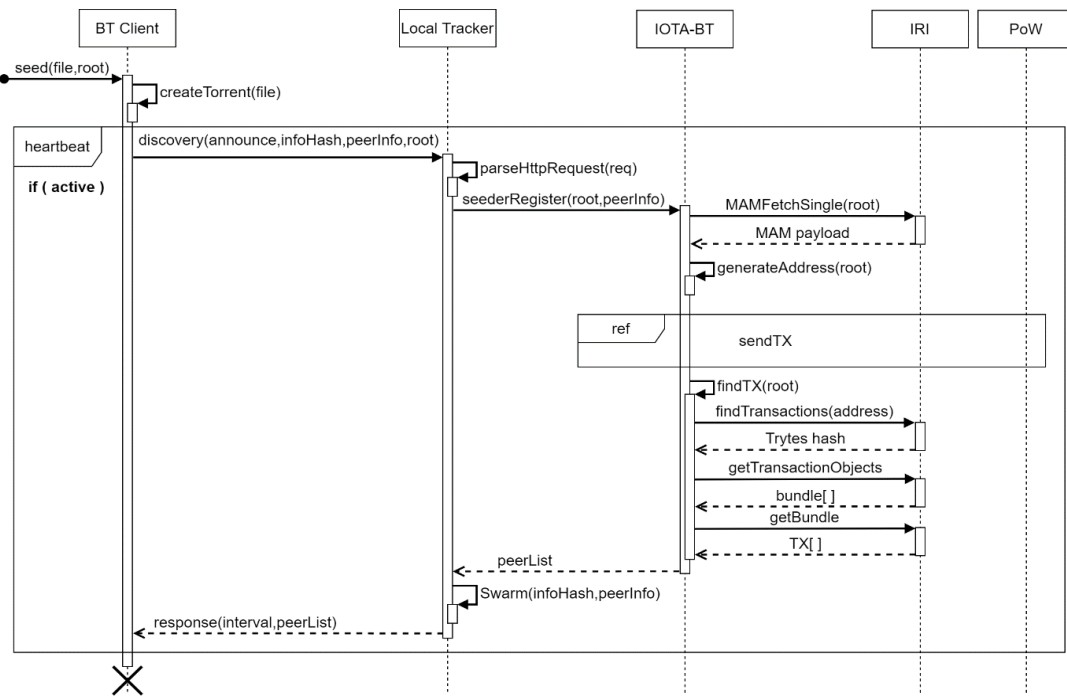

**Figure 6.** Seeding process in IOTA-BT.

When the IOTA-BT client obtains the MAM Root of a shared file, it can download the file. The download process is shown in Figure 7. (1) Use the root to query MAM transactions and obtain seed files recorded in transactions. (2) Determine whether to proceed with the download process by comparing the size of the file it owns with that recorded in the seed file. (3) If the previous step determines to proceed with the download process, the client performs the heartbeat process to obtain the peer list. (4) Send file request inquiries to each node recorded in the peer list and ask whether the node has the specified file block. If the node has the file block, it sends the file block to the client. After receiving all file blocks, the client assembles the file blocks to form a complete file.

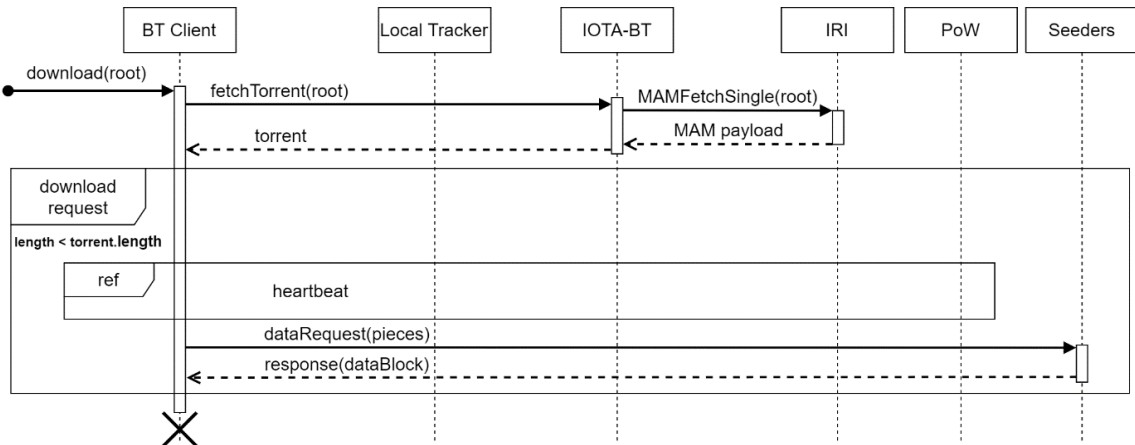

**Figure 7.** Download process in IOTA-BT.

To filter invalid node information, IOTA-BT sets an expiration date for node information. As a result, outdated information becomes invalid over time. The client must register periodically to make others know it is alive to provide file transfer services. However, the continuous registration of the same address also increases the cost of searching for transactions. Therefore, this research adopts the

time difference registration method proposed by Tsung-Yi Tang [35]. The principle of the method is shown in Figure 8.

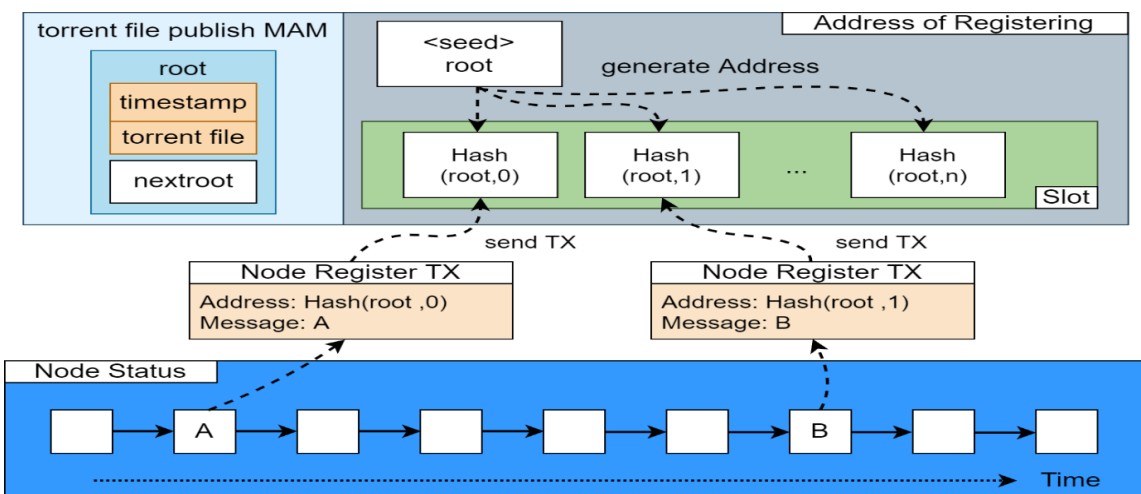

**Figure 8.** Time difference registration method.

IOTA's native address generation is to generate a set of addresses by hashing a seed with a length of 81 tryte and a positive integer index. Each address can be used repeatedly if there is no amount of money transfer behavior, and a single address can contain multiple transactions. If there is a transfer behavior, the balance will be transferred to the address of the next index for storage. In this study, the root of the MAM message recorded in the seed file was used as the seed of the generated address, and the time difference between the current time and the release time of the MAM message was used as the index to form a time difference address for node registration. As a consequence, the newest node information can be registered at the latest time address. When a client searches for the seeder of a specific file, it can efficiently obtain the newest node information in the latest time slot. For example, after obtaining the root of the MAM message, a client registers at time point A and packs its node information into a message, and sends it to a designated address, which is generated as follows: (1) calculate the gap between the time stamp in the MAM message and time point A. (2) Divide the gap by the length of a time slot to obtain the slot index value for storing the message. (3) Perform address generation by using the slot index value and MAM message root. The previous process will be executed again when registration is performed next time (time point B). When clients need to know the latest information, they only need to query in the latest slot to obtain the currently alive node information. Therefore, it is useful for filtering out the expired node information and reducing unnecessary data search time.

## 4. Performance Evaluation

We have evaluated the performance of IOTA-BT on IOTA public chain in this paper. In an experimental environment, we set up three IRI nodes located on Tokyo Japan, London U.K, and Los Angeles U.S by using virtual machines from the Google Cloud Platform, and built up one IRI node by one PC in our lab. These four nodes were used for transaction listening and broadcasting. The communication between clients and IRI nodes relied on HTTP-ping. Besides, the IRI nodes adopted ZMQ (zero message queue) to accept and listen to transactions. According to our estimation, the average cost of HTTP-ping was 0.0473 ms, and the average time cost of transaction broadcasting was 425.656 ms. On the other hand, the client node used in this performance evaluation was one PC that has one Intel i7-8700 CPU and one Nvidia 1080Ti graphic cards and 32GB RAM. Its job was mainly to do transaction storing, broadcasting, and PoW. On the other hand, IOTA-BT API was deployed by OpenFass, and the client performed transaction packing and transferring. The time interval of

heartbeats was set as 3 min. Our experiments were focused on the costs of releasing transactions and torrent files and the cost of obtaining seeder information. Our experimental results are discussed as follows.

First, publishing an IOTA transaction needs five steps: bundle, getTransactionToApprove (GATT), attachToTangle (ATT), storeTransactions (ST), and broadcastTransactions (BT). In this experiment, a bundle had only one transaction, and the depth of getTransactionToApprove was 1; the mwm parameter of attachToTangle in the mainnet was 14. The client used Nvidia GTX 1080Ti to perform PoW.

The experimental result is shown in Figure 9. In the figure, BUNDLE means the time of packing a transaction. GATT is the time spent on tip select by the MCMC algorithm. ATT is the cost of PoW. ST is the cost of storing a transaction on IRI. BT is the broadcasting time of IRI. The experimental result demonstrates that the biggest cost of publishing a transaction is spent on ATT (i.e., PoW), and the other costs are almost negligible. Moreover, the getTransactionToApprove function of IOTA-BT API can change the traversal depth of MCMC algorithm by the depth parameter. The influence of traversal depth on the cost of the tip select is depicted in Figure 10. The traversal depth has influence if the transaction is bound to unreliable tips. The shorter the traversal depth, the more easily bound the transaction is to unreliable tips. However, the IOTA-BT transaction is zero-fee transactions. The transaction verification does not affect the operation of IOTA-BT. Therefore, IOTA-BT set the traversal depth as 1 for reducing the cost of transaction publishing.

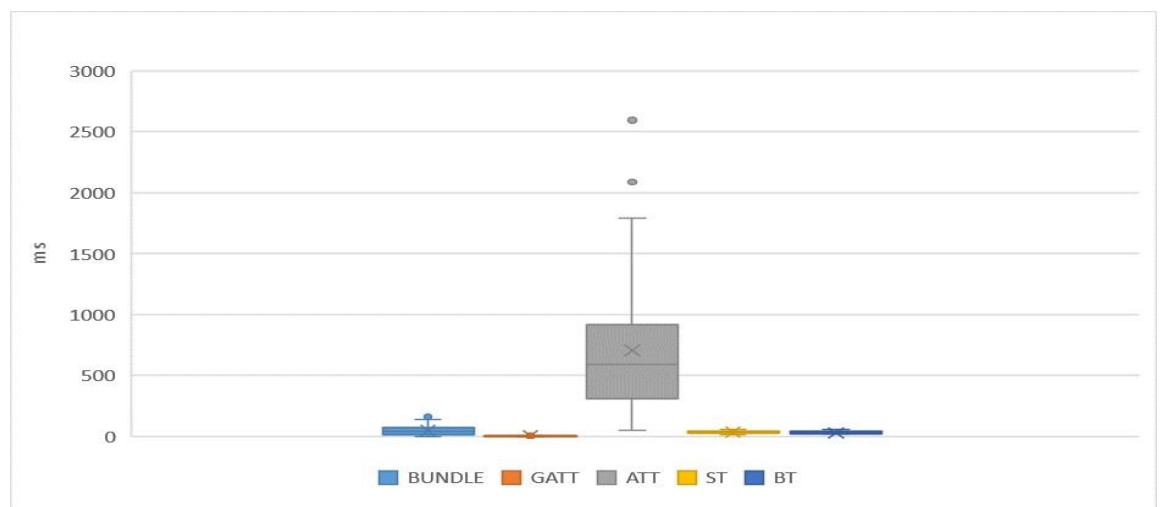

**Figure 9.** Cost of transaction publishing.

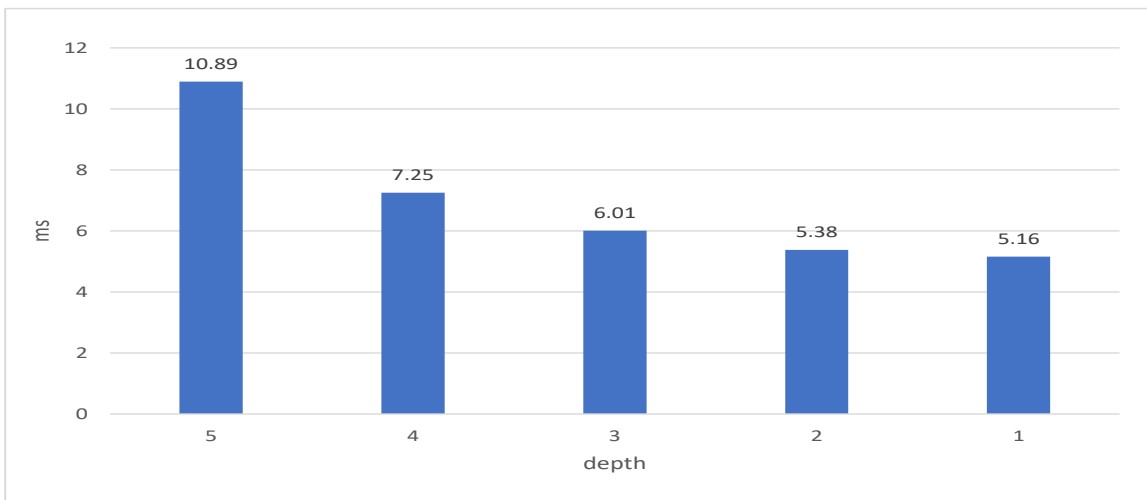

**Figure 10.** The cost of tip select by different traversal depths.

On the other hand, the IOTA mainnet requires the mwm parameter of PoW must be equal to or larger than 14. We evaluated the cost of PoW by CPU and GPU, respectively. IOTA used the original API of IOTA to execute PoW by CPU while using the ccurl library of OpenCL to do PoW by GPU. The transaction content was statically assigned. Figure 11 shows that the average time of publishing 1TX and 5TX 200 times. Compared with the i7-8700 CPU, using a GPU effectively reduces the 10% of the PoW cost.

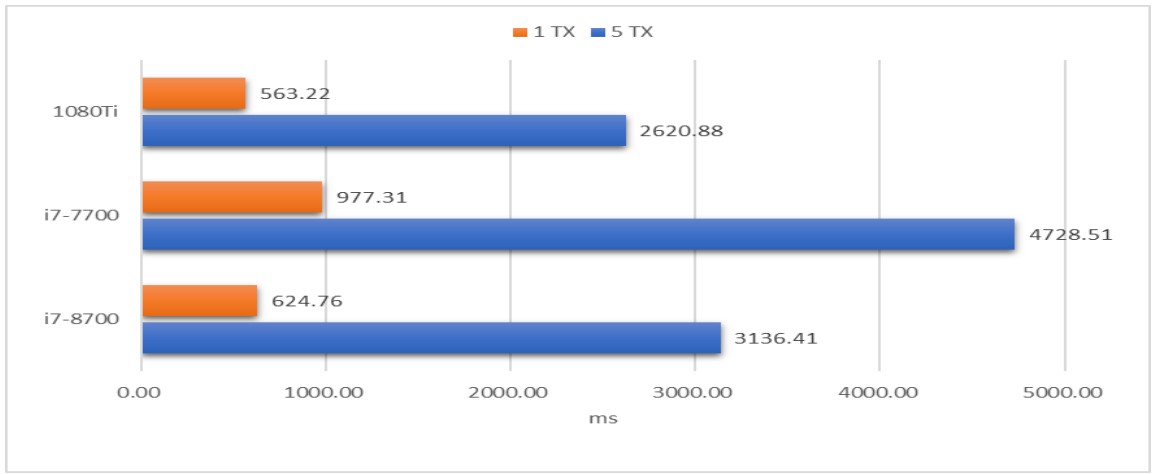

**Figure 11.** Cost of Proof-of-Work (PoW).

In IOTA-BT, the file owner must publish a torrent file to IOTA in the form of MAM, so that users can query it through the MAM Root. We used CPU and GPU to publish torrent files 100 times and calculates the average time of publishing torrent files. The result is shown in Figure 12, which includes MAM address generation and transaction initiation. Since the GPU spends less time on PoW when initiating a transaction, it can reduce the time spent on transaction publishing. Although the time cost of transaction publishing is about 6~8 s, it is acceptable for improving the security of file sharing in BitTorrent.

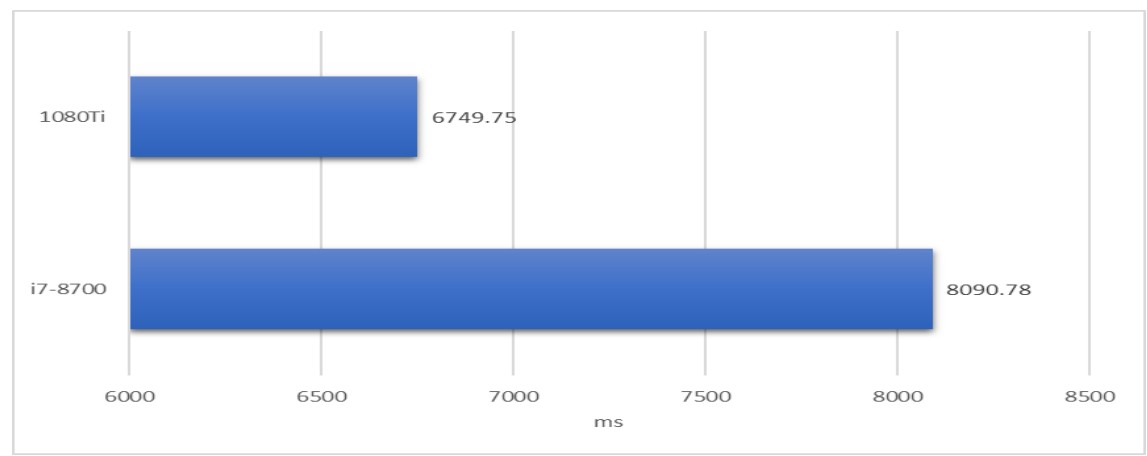

**Figure 12.** Cost of publishing a Masked Authenticated Message (MAM) transaction.

When the client joins into IOTA-BT, it needs to register its node no matter what it wants to do. The steps of node registration consist of the following steps: (1) Perform the MAM query to obtain infoHash, the file block list, file release time, and other information. (2) Use the current time to generate the registered address based on the time difference. (3) Send its node information in the form of general transactions. In Figure 13, Fetch MAM represents the time of querying MAM content to obtain the torrent file and release time, Generate TX is the time required to generate the transaction address,

and Send TX is the time required for sending a new transaction by verifying two other transactions. The experimental result shows that it takes only 1.6 s to register a node, and the performance difference between the CPU and GPU is only about 200 ms, mainly due to the time difference of PoW.

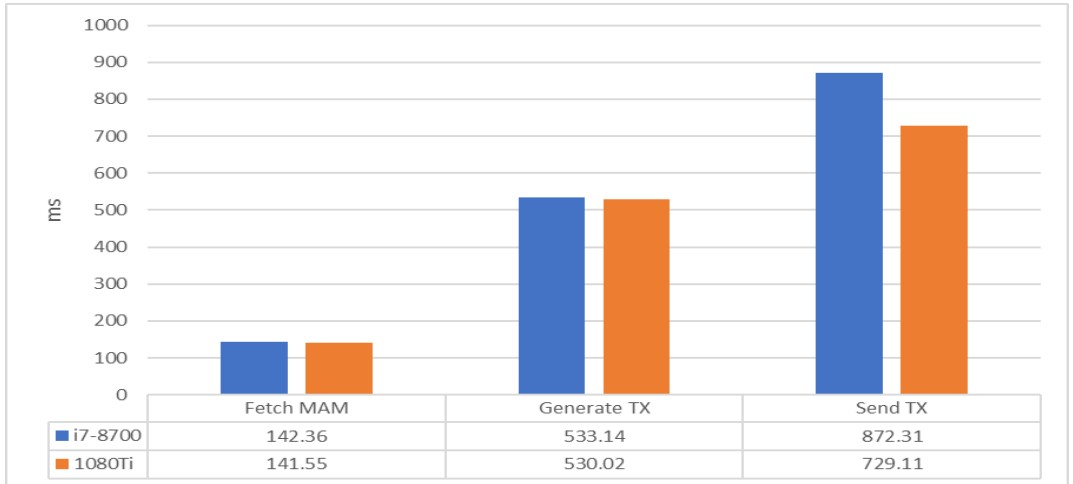

| | Fetch MAM | Generate TX | Send TX |
|---|---|---|---|
| ■ i7-8700 | 142.36 | 533.14 | 872.31 |
| ■ 1080Ti | 141.55 | 530.02 | 729.11 |

**Figure 13.** Registration cost.

For obtaining node information fast, IOTA-BT adopts a time difference node registration mechanism to improve its efficiency. Assume that every six minutes is regarded as a time slot, and the node sends a heartbeat every three minutes. The node will generate at most 480 transactions for registering the same service in one day. If a service has only one available node, the user can query the available node information of this service from at most two data. This study measures the time spent on each stage of the registration process. PoW is not necessary for generating addresses and finding transactions. Therefore, this experiment is measured by using the CPU, and the result is shown in Figure 14. In the figure, Generate Address is the time of generating the registered address, Find TX is the time used to find the registered address, Get TX is the time spent on getting the transaction, and Decode TX is the time of parsing the transaction content. In the process, Generate Address takes the most time, but it is almost constant. Secondly, the time spent on Decode TX, Find TX and Get TX depends on the current load status of IRI. It can be seen from the results that Decode TX time is most affected by the number of transactions. IOTA-BT uses a time difference registration mechanism to filter outdated information, which can effectively reduce the amount of useless node information and improve the efficiency of fetching node information.

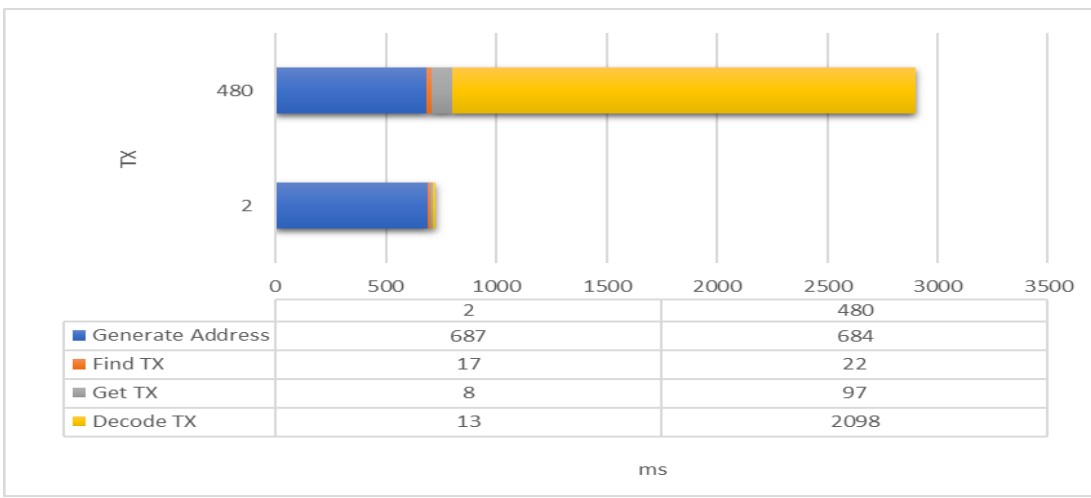

| | 2 | 480 |
|---|---|---|
| ■ Generate Address | 687 | 684 |
| ■ Find TX | 17 | 22 |
| ■ Get TX | 8 | 97 |
| ■ Decode TX | 13 | 2098 |

**Figure 14.** Cost of obtaining node information.

Finally, we attempted to compare the performance difference between IOTA-BT and traditional BT. However, the cost of PoW is dependent on the computational power of used resources. Therefore, we focused only on the cost of getting the peer list here. We set up four BT nodes in Taiwan, Tokyo, London, and Los Angeles (LA), respectively. The node in Taiwan played the tracker server in the BT framework and the seed node for a shared file. In contrast, the other nodes fetched the peer list for downloading the shared file from the Taiwan node. We measured how much it cost the other nodes to acquire information about the Taiwan node. The measurement result is shown in Figure 15. The node in Tokyo spent the least time while the node in London spent the most time. The average cost of getting the peer list is 310 ms. Compared with Figure 14, the time difference is only 415 ms. It is ignorable for users compared with the time of file downloading. As previously discussed, IOTA-BT can support higher security for P2P file sharing than traditional BT with a few extra costs.

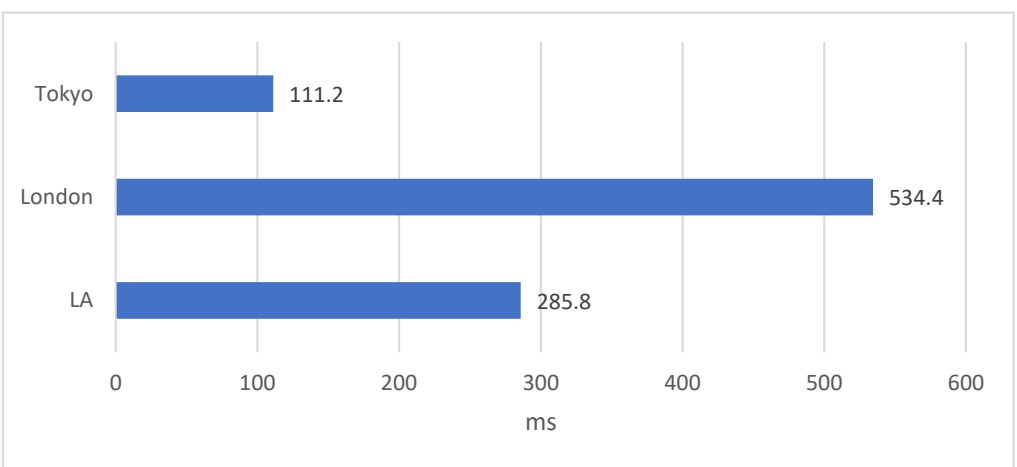

**Figure 15.** Cost of getting a peer list in the BT network.

## 5. Conclusions and Future Work

In this paper, we have successfully developed an IOTA-based BitTorrent system based on distributed ledger technology. IOTA-BT enables the original BT framework to be executable on the IOTA mainnet to support a P2P file-sharing service with higher security, availability, and public access. The experimental result shows that the client takes only 6~8 s to release a seed file in the IOTA mainnet, and then other clients can query the detail of the seed file through any IOTA mainnet node within one second. Besides, a node takes only 1.4 and 0.7 s to register its heartbeat and obtain the peer list, respectively. It is almost negligible for users because the time interval of issuing a heartbeat in the BT network is 10 min. Therefore, it is recommended that the time interval length of issuing a heartbeat can be further shortened to increase the frequency of information exchange and reduce the time of file downloading. On the other hand, the experiment result also shows that IOTA-BT can indeed effectively reduce the number of unnecessary transaction searches and decoding. Finally, the efficiency of IOTA-BT is indeed acceptable for P2P file sharing.

IOTA recently has released the Coordicide white paper, and it is ready to move towards version 2.0. Its goal is to achieve true decentralization, abolish the PoW mechanism, and realize smart contracts with more application value. We will exploit IOTA 2.0 smart contracts to make IOTA-BT evolve into a P2P file trading system in the future. Additionally, service discovery is a critical issue for fog computing. Since the resources of fog computing are distributed to different network levels or organizations, a central service discovery mechanism is not practical and realistic. We will develop a distributed and public service discovery mechanism for fog computing by referencing the framework of IOTA-BT.

**Author Contributions:** Conceptualization, T.-Y.L.; Data curation, L.-Y.H.; Funding acquisition, T.-Y.L.; Investigation, T.-Y.L.; Methodology, T.-Y.T. and T.-Y.L.; Project administration, T.-Y.L.; Resources, L.-Y.H.; Software, L.-Y.H. and T.-Y.T.; Supervision, T.-Y.L.; Validation, L.-Y.H.; Visualization, L.-Y.H.; Writing—review & editing, T.-Y.L. All authors have read and agreed to the published version of the manuscript.

**Funding:** This research was funded by the Ministry of Science and Technology in Taiwan grant number [MOST 108-2221-E-992-030].

**Acknowledgments:** We would like to thank the support of the Ministry of Science and Technology in Taiwan on this study under the project number: MOST 108-2221-E-992-030.

**Conflicts of Interest:** The authors declare no conflict of interest.

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
