# Peer review of "IOTA-BT: A P2P File-Sharing System Based on IOTA"

_electronics, doi:10.3390/electronics9101610_

Round 1
Reviewer 1 Report
The authors present a file sharing system based on IOTA that tries to overcome the limitations of the already existing solutions, such as BitTorrent. The usage of IOTA is justified, among the other, by the fact that it does not require a central tracker, which can be the target of several attacks that can undermine the stability of the network.
After an introduction about BitTorrent structure an mechanism and IOTA, authors introduce their proposal (IOTA-BT) in Section 3.
There are some minor english issues such as in page 2: attacks on the blockchain are not economical -> not cheap.
Figures 4, 8 and 9 should be of better quality.
It would be interesting to provide the link to the implementation in the paper in order to try the code. And moreover would be interesting to provide a more in depth analysis of the related works.
Overall, the paper present an interesting approach and is clear, so i suggest to accept it after some minor fixing.
Author Response
Response to the first reviewer’s comments
Q1:>There are some minor English issues such as on page 2: attacks on the blockchain are not economical -> not cheap.
Response: Thanks for the comment. We have revised English in this paper with the help of a colleague who is a native speaker in English.
Q2> Figures 4, 8, and 9 should be of better quality.
Response: We have increased the quality of these figures by using a higher resolution.
Q3:> It would be interesting to provide the link to the implementation in the paper to try the code. And moreover would be interesting to provide a more in depth analysis of the related works.
Response: We are glad that the reviewer is interested in our project. We will release and share the software of the IOTA-BT in public after our paper is published. We need some time to pack the source codes into a Github package and write a document to be a user guide of the proposed system. On the other hand, we have compared IOTA with Ethereum to explain why we choose IOTA instead of Ethereum for implementing the BT framework. Furthermore, we have compared the performance of IOTA-BT with that of traditional BT on page 14 to show that IOTA-BT indeed can provide higher security for P2P sharing while spending a few extra costs.

Reviewer 2 Report
This is an interesting article on BitTorrent (BT) peer-to-peer file sharing system and security threats including Sybil and Eclipses. The authors proposed a method called IOTA-BT by integrating BitTorrent with IOTA in this paper. The paper revealed acceptable results after evaluating the proposed method.
Issues:
For a journal paper, the literature review is too superficial. The quality of references is good; however, the paper needs to provide critical discussion with reference to literature sources using appropriate discussion metrics.
The paper needs appropriate structure. Currently, everything is tucked under Background. It could follow standard organisation providing details of research methodology. How was the study conduction with results and discussion section.
The result section lacks details and should include critical discussion and comparison with other methods with reference to literature sources. All figures presented in the paper should be discussed and referenced in text. For example what is the purpose figure 11.
The conclusion section should discussion recommendations and future direction.
Improvement is needed in overall written expression of the manuscript. There are grammatical and language mistakes which should be rectified to make the study more appealing and effective. The paper needs a through language review before publication.
Author Response
Responses to the second reviewer’s comment
Comment 1: For a journal paper, the literature review is too superficial. The quality of references is good; however, the paper needs to provide critical discussion with reference to literature sources using appropriate discussion metrics.
Response: As we know, this paper is the first work of applying blockchain to BT. So, we add the comparison between IOTA and Ethereum for the implementation of BT in page 6.
Comment 2: The paper needs appropriate structure. Currently, everything is tucked under Background. It could follow standard organisation providing details of research methodology. How was the study conduction with results and discussion section.
Response: Thanks for the comment. We have provided the necessary background for our target (i.e., BT) and method (i.e., IOTA) in section 2. We also compared IOTA with Ethereum to explain why we choose IOTA instead of Ethereum for implementing the BT framework on page 6.
Comment 3: The result section lacks details and should include critical discussion and comparison with other methods with reference to literature sources. All figures presented in the paper should be discussed and referenced in text. For example what is the purpose figure 11.
Response: We have revised the performance section by discussing and comparing the performance of IOTA-BT with that of traditional BT on page 14. On the other hand, we also confirm that all figures are referenced and discussed in the main text after paper revision.
Comment 4: The conclusion section should discuss recommendations and future direction.
Response: We have discussed our recommendations and future work in the conclusion section of this paper as shown on page 15.
Comment 5: Improvement is needed in overall written expression of the manuscript. There are grammatical and language mistakes which should be rectified to make the study more appealing and effective. The paper needs a through language review before publication.
Response: Thanks for the comment. We have revised English in this paper with a colleague who is a native speaker in English.
